# ACCELERATED SPARSE RECOVERY UNDER STRUCTURED MEASUREMENTS

## ABSTRACT

Extensive work on compressed sensing has yielded a rich collection of sparse recovery algorithms, each making different tradeoffs between recovery condition and computational efficiency. In this paper, we propose a unified framework for accelerating various existing sparse recovery algorithms without sacrificing recovery guarantees by exploiting structure in the measurement matrix. Unlike fast algorithms that are specific to particular choices of measurement matrices where the columns are Fourier or wavelet filters for example, the proposed approach works on a broad range of measurement matrices that satisfy a particular property. We precisely characterize this property, which quantifies how easy it is to accelerate sparse recovery for the measurement matrix in question. We also derive the time complexity of the accelerated algorithm, which is sublinear in the signal length in each iteration. Moreover, we present experimental results on real world data that demonstrate the effectiveness of the proposed approach in practice.

## 1 INTRODUCTION

Natural data in its original form often contains much redundant information. In many domains, data consists of multiple different (possibly noisy) linear measurements of the same latent signal, and so redundancy implies that the amount of information encoded by the signal is often small relative to its length. Because taking measurements in the real world is expensive, one of the goals of signal processing is to reconstruct the latent signal with as few measurements as possible. Compressed sensing (Donoho, 2006; Candes et al., 2006) refers to the idea of exploiting the redundancy to reconstruct the signal from highly incomplete measurements, where the number of measurements is much smaller than the length of the signal.

Mathematically, (discrete-time) signals that give rise to redundant data take the form of vectors whose entries decay quickly when sorted by magnitude in decreasing order. As such, they can be approximated by sparse vectors. Compressed sensing works by encouraging sparsity in the recovered signal, while ensuring it is consistent with the observed measurements. Formally, its goal is to solve the following optimization problem, which is known as the sparse recovery problem:

$$\min_{\mathbf{x} \in \mathbb{R}^n} \|\mathbf{x}\|_0 \text{ s.t. } A\mathbf{x} = \mathbf{y}$$

where $A \in \mathbb{R}^{m \times n}$ is the measurement matrix, $\mathbf{y} \in \mathbb{R}^m$ is the vector of observed measurements and $\mathbf{x} \in \mathbb{R}^n$ is a vector representing a possible recovered signal. $\|\cdot\|_0$ denotes the $\ell_0$ "norm" [1], which returns the number of non-zero entries of a vector. Each row of $A$ represents a particular way of measuring the signal $\mathbf{x}$. The number of measurements, $m$, is usually much smaller than the length of the signal, $n$.

This problem is known to be NP hard in general (Natarajan, 1995), and so work in sparse recovery focuses on developing algorithms that can efficiently recover any possible sparse latent signal for particular classes of measurement matrices $A$. One way of characterizing the hardness of $A$ is via its restricted isometry constant (Candes & Tao, 2005). More specifically, a matrix has a $k$-restricted isometry constant $\delta_k$ if $(1 - \delta_k) \|\mathbf{x}\|_2^2 \le \|A\mathbf{x}\|_2^2 \le (1 + \delta_k) \|\mathbf{x}\|_2^2$ for all $k$-sparse vectors $\mathbf{x}$, that is, vectors with at most $k$ non-zero entries. Intuitively, a small $k$-restricted isometry constant means

---

[1] Quotation marks are due to the fact that $\|\alpha \mathbf{x}\|_0 \ne |\alpha| \|\mathbf{x}\|_0$ in general.

that any set of $k$ columns of the matrix is nearly orthonormal, since restricted isometry implies that $\left\| A_{.J}^T A_{.J} - \mathbf{I} \right\|_2 \leq \delta_k$ for any $J \subseteq \{1, \ldots, n\}$ such that $|J| \leq k$, where $A_{.J}$ denotes the submatrix of $A$ consisting of columns indexed by $J$. (Candes & Tao, 2005) showed that as long as $\delta_k + \delta_{2k} + \delta_{3k} < 0.25$, any $k$-sparse vector can be recovered efficiently.

Various algorithms have been developed for sparse recovery, each of which trades off recovery guarantees, that is, the conditions under which the algorithm succeeds, for computational efficiency. (See the next section for details.) In this paper, we explore an orthogonal approach that leverages structure in measurement matrices to improve the computational efficiency of various existing algorithms. While there have been prior methods along this direction, they either require carefully chosen measurement matrices (like columns consisting of Fourier or wavelet filters (Gilbert et al., 2002; 2005), or other specially designed constructions (Gilbert et al., 2006; 2007)) or compromise on recovery guarantees (Jain et al., 2011). In contrast, we systematically characterize all measurement matrices in terms of their ability to support efficient sparse recovery and quantify it using a single number, known as the intrinsic dimensionality. We show that acceleration is possible as long as intrinsic dimensionality is relatively small and precisely delineate the dependence of running time on this number.

Many sparse recovery algorithms multiply an tall matrix with a low-dimensional vector, which results in a high-dimensional vector, only to discard most of the elements in the vector later. In the case of greedy pursuit algorithms, given a sparse vector $\mathbf{x}$, $A^T$ is multiplied with the residual vector $A\mathbf{x} - \mathbf{y}$ to find the next coordinates to add to the support. Because the added coordinates correspond to the elements of the resulting vector with the largest magnitudes, most of the elements with small magnitudes have no effect on the algorithm and their values are discarded. Similarly, in the case of iterative shrinkage algorithms, $A^T$ is multiplied with the residual when computing the gradient, which is added to the current iterate and transformed by a shrinkage operator. Because many shrinkage operators zero out elements with small magnitudes, the values of those elements are effectively discarded. In both cases, because each element is derived from the result of an expensive inner product computation that takes $O(mn)$ time, and there are many elements whose values are discarded, there is a lot of wasted computation, which could potentially be saved if we know *a priori* which elements will be discarded. Of course, the challenge is that we don't know a priori which elements will be discarded, since that depends on their values, which we don't know unless we compute them. Is there any way around this? Somewhat surprisingly, the answer is yes. By leveraging the fact that the matrix residuals are multiplied with is always $A^T$ over all iterations, it is possible to preprocess $A^T$ ahead of time so that we can quickly identify the elements in $A^T\mathbf{v}$ that would have the largest magnitudes for any $\mathbf{v}$ without multiplying all rows of $A^T$ with $\mathbf{v}$. Hence, by using this technique, the time complexity per iteration becomes *sublinear* in the length of the latent signal $n$. While one tempting way of accelerating the identification of the support is to reduce the problem to hashing; however, as shown by (Jain et al., 2011), such an approach would result in a significant degradation in recovery guarantees. In contrast, our technique is guaranteed to preserve recovery guarantees. Empirically, we observe various algorithms accelerated using this technique achieve significant speedups compared to the vanilla versions.

## 2 BACKGROUND

There are four major types of methods for sparse recovery: basis pursuit, greedy pursuit, iterative shrinkage and iterative reweighting. Basis pursuit (Chen et al., 2001) replaces the $\ell_0$ "norm" with $\ell_1$ norm, which is convex and makes the problem much easier to solve. The $\ell_1$ problem can be then rewritten as a linear program and solved using simplex or interior-point methods. Greedy pursuit algorithms maintain an estimated support set of the latent signal, which consists of the coordinates of the latent signal the algorithm hypothesizes to be non-zero. In each iteration, they perform least squares on the submatrix of $A$ consisting of columns in the support and incrementally add or remove coordinates from the support based on the current residual. Examples of algorithms in this category include orthogonal matching pursuit (OMP) (Pati et al., 1993; Tropp & Gilbert, 2007), stagewise OMP (StOMP) (Donoho et al., 2012), regularized OMP (ROMP) (Needell & Vershynin, 2009), compressive sensing matching pursuit (CoSaMP) (Needell & Tropp, 2009), subspace pursuit (SP) (Dai & Milenkovic, 2009), hard thresholding pursuit (Foucart, 2011) and OMP with replacement (OMPR) (Jain et al., 2011). Iterative shrinkage algorithms solve a noisy version of the original

problem and possibly relax it by considering $q > 0$: [2]

$$\min_{\mathbf{x} \in \mathbb{R}^n} \|\mathbf{x}\|_q^q \ \text{ s.t. } \ \|A\mathbf{x} - \mathbf{y}\|_2^2 \leq \tau$$

They first convert the problem into the following related unconstrained problem, which is equivalent to the above for some unknown value of $\tau$:

$$\min_{\mathbf{x} \in \mathbb{R}^n} \|A\mathbf{x} - \mathbf{y}\|_2^2 + \lambda \|\mathbf{x}\|_q^q$$

They then iteratively descend along the gradient of $\|A\mathbf{x} - \mathbf{y}\|_2^2$ w.r.t. $\mathbf{x}$ (known as a Landweber iteration) and then applies a shrinkage operator to each coordinate of the iterate independently. The shrinkage operator varies for different algorithms and depends on the value of $q$ that the algorithm optimizes for, and either reduces or maintains the magnitude of each coordinate. Examples of algorithms in this category include iterative soft thresholding (Donoho, 1995; Daubechies et al., 2004; Maleki & Donoho, 2010), iterative hard thresholding (Blumensath & Davies, 2009), iterative half thresholding (Xu et al., 2012), adaptive iterative soft and hard thresholding (Wang et al., 2015), accelerated hard thresholding (Cevher, 2011). Like iterative shrinkage algorithms, iterative reweighting algorithms convert the original problem into the unconstrained version; unlike iterative shrinkage, they convert the non-convex $\ell_q$ norm into convex $\ell_2$ or $\ell_1$ norms by setting a weight on each coordinate and solve a sequence of weighted $\ell_2$ or $\ell_1$ regression problems.

Each of these methods make different tradeoffs between recovery condition and computational efficiency. Basis pursuit has excellent recovery guarantees and is able to recover all $k$-sparse vectors if $\delta_{2k} < 0.707$ (Cai & Zhang, 2014). Unfortunately, it becomes computationally expensive in high dimensions, where the length of the signal could be on the order of the millions or more. In terms of computational complexity, no known linear programming algorithms achieve a *strongly* polynomial running time, that is polynomial only in the number of variables, which correspond to $m + n$ in the original sparse recovery problem, and number of constraints without any dependence on condition numbers of the program. It is currently not known if small restricted isometry constants imply the conditional numbers are polynomial in $m$ or $n$.

On the other hand, OMP is very computationally efficient both in theory and in practice. To recover a $k$-sparse vector, it only needs to perform $k$ iterations, where in each iteration it performs a matrix multiplication with $A^T$ and solves a least squares problem on a $m \times k$ submatrix of $A$. Hence, the algorithm is guaranteed to finish in strongly polynomial time, regardless of what $A$ is. Unfortunately, OMP has much weaker recovery guarantees and is only known to be able to recover $k$-sparse vectors correctly if $\delta_{13k} < 1/6$ (Foucart & Rauhut, 2013), which is a much more stringent condition that that is required by basis pursuit. Other greedy pursuit algorithms achieve better recovery guarantees: ROMP requires $\delta_{3k} < 0.2$ (Needell & Vershynin, 2009), CoSaMP requires $\delta_{4k} < 0.384$ (Foucart, 2012), SP requires $\delta_{3k} < 0.35$ (Jain et al., 2011) and OMPR requires $\delta_{2k} < 0.499$ (Jain et al., 2011). However, to date, none have been able to achieve comparable recovery guarantees as basis pursuit. Similarly, iterative shrinkage algorithms are also more computationally efficient than basis pursuit, but have weaker recovery guarantees. For example, the best known guarantees for iterative hard thresholding are $\delta_{3k} < 0.577$ (Foucart & Rauhut, 2013) and $\delta_{3k+1} < 0.618$ (Wang et al., 2015).

## 3 METHOD

We consider two examples of greedy pursuit and iterative shrinkage algorithms, compressive sensing matching pursuit (CoSaMP) and adaptive iterative hard thresholding (AIHT).

We first make use of the fact that in sparse recovery problems, the columns of $A$ can be normalized without loss of generality. More formally,

**Fact 1.** *For any invertible diagonal matrix $D \in \mathbb{R}^{n \times n}$, the following optimization problems are equivalent:*

$$\min_{\mathbf{x} \in \mathbb{R}^n} \|\mathbf{x}\|_0 \ \text{ s.t. } \ \|A\mathbf{x} - \mathbf{y}\|_2^2 \leq \tau \tag{1}$$

$$\min_{\mathbf{x} \in \mathbb{R}^n} \|\mathbf{x}\|_0 \ \text{ s.t. } \ \|AD\mathbf{x} - \mathbf{y}\|_2^2 \leq \tau \tag{2}$$

---

[2] where we define $\|\mathbf{x}\|_0^0$ to be $\|\mathbf{x}\|_0$ for notational convenience

*Proof.* Define $\tilde{\mathbf{x}} := D^{-1}\mathbf{x}$, and so $\mathbf{x} = D\tilde{\mathbf{x}}$. We substitute this into problem (1), and obtain:

$$\min_{D\tilde{\mathbf{x}} \in \mathbb{R}^n} \|D\tilde{\mathbf{x}}\|_0 \text{ s.t. } \|AD\tilde{\mathbf{x}} - \mathbf{y}\|_2^2 \leq \tau$$

Because $D$ is diagonal and invertible, an element of $D\tilde{\mathbf{x}}$ is zero if and only if the corresponding element of $\tilde{\mathbf{x}}$ is zero. So, the number of zeros in $D\tilde{\mathbf{x}}$ is the same number of zeros in $\tilde{\mathbf{x}}$. Hence, $\|D\tilde{\mathbf{x}}\|_0 = \|\tilde{\mathbf{x}}\|_0$. The set the optimization is performed over is $D^{-1}(\mathbb{R}^n)$, which is the same as $\mathbb{R}^n$ because $D$ is invertible. $\square$

If we choose $D_{jj}$ to be $1/\|A_{\cdot j}\|_2$ and define $A' := AD$, then each column of $A'$ would have unit norm. Therefore, we will henceforth assume that $A$ is column-normalized.

Our goal is to speed up the computation of $A^T\left(A\mathbf{x}^{(t-1)} - \mathbf{y}\right)$ by taking advantage of the fact that most of the elements in the result will be discarded when the new support is computed or when the shrinkage operator is applied. We first consider CoSaMP, which is delineated in Algorithm 1. In CoSaMP, the result of $A^T\left(A\mathbf{x}^{(t-1)} - \mathbf{y}\right)$ is stored in $\mathbf{z}^{(t)}$, which is then used to find the set $U$, consisting of new coordinates to add the support. To compute $U$, we need to know which elements of $\mathbf{z}^{(t)}$ are the largest in magnitude; other than this step, $\mathbf{z}^{(t)}$ is not used anywhere else. So, it suffices to identify which elements of $\left|\mathbf{z}^{(t)}\right|$ are largest without necessarily computing all elements of $\mathbf{z}^{(t)}$ explicitly.

---

**Algorithm 1** Compressive Sensing Matching Pursuit (CoSaMP)

---

**Require:** Measurement matrix $A$, observed measurements vector $\mathbf{y}$ and sparsity level $k$
  $\mathbf{x}^{(0)} \leftarrow \mathbf{0}$
  $S \leftarrow \emptyset$
  **for** $t = 1$ **to** $T$ **do**
    $\mathbf{z}^{(t)} \leftarrow A^T\left(A\mathbf{x}^{(t-1)} - \mathbf{y}\right)$
    $U \leftarrow$ indices of the $2k$ largest elements of $\mathbf{z}^{(t)}$ in magnitude
    $\tilde{S} \leftarrow U \cup S$
    $\left.\tilde{\mathbf{x}}^{(t)}\right|_{\tilde{S}} \leftarrow \left(\left(A|_{\cdot\tilde{S}}\right)^T A|_{\cdot\tilde{S}}\right)^{-1} A|_{\cdot\tilde{S}}^T \mathbf{y}$
    $\left.\tilde{\mathbf{x}}^{(t)}\right|_{\tilde{S}^c} \leftarrow \mathbf{0}$
    $S \leftarrow$ indices of the $k$ largest elements of $\tilde{\mathbf{x}}^{(t)}$ in magnitude
    $\left.\mathbf{x}^{(t)}\right|_S \leftarrow \left.\tilde{\mathbf{x}}^{(t)}\right|_S$
    $\left.\mathbf{x}^{(t)}\right|_{S^c} \leftarrow \mathbf{0}$
  **end for**
  **return** $\mathbf{x}^{(T)}$

---

Consider the $j$th element of $\mathbf{z}^{(t)}$, which we will denote as $z_j^{(t)}$. It can be written as follows:

$$z_j^{(t)} = \langle A_{\cdot j}, A\mathbf{x}^{(t-1)} - \mathbf{y}\rangle$$

Therefore, to find the largest elements of $\left|\mathbf{z}^{(t)}\right|$, we need to find the $A_{\cdot j}$'s that have the highest inner products with $A\mathbf{x}^{(t-1)} - \mathbf{y}$ in absolute value. We make use of the following fact to do so:

**Fact 2.** *Let $\mathcal{D}$ be a set of vectors and $S \subseteq \mathcal{D}$ be the subset of vectors that attain the $\tilde{k}$ highest inner products with $\mathbf{w}$ in absolute value. Let $S_+$ and $S_-$ be subsets of vectors that attain the $\tilde{k}$ highest inner products with $\mathbf{w}$ and $-\mathbf{w}$ respectively (not in absolute value). Then $S \subseteq (S_+ \cup S_-)$.*

*Proof.* Consider a partitioning of $\mathcal{D}$ into two disjoint subsets, $\mathcal{D}_+$ and $\mathcal{D}_-$, which consist vectors in $\mathcal{D}$ that have positive and non-positive inner products with $\mathbf{w}$ respectively.

For any $\mathbf{v} \in S$, either $\langle \mathbf{v}, \mathbf{w}\rangle > 0$ or $\langle \mathbf{v}, \mathbf{w}\rangle \leq 0$. If $\langle \mathbf{v}, \mathbf{w}\rangle > 0$, then only the other vectors in $S$ can have larger inner products with $\mathbf{w}$ than $\mathbf{v}$. There are at most $k - 1$ of these vectors, and so $\mathbf{v} \in S_+$. Similarly, if on the other hand $\langle \mathbf{v}, \mathbf{w}\rangle \leq 0$, then only the other vectors in $S$ can have smaller inner products with $\mathbf{w}$ than $\mathbf{v}$. There are at most $k - 1$ of these vectors, which implies that $\mathbf{v} \in S_-$. This completes the proof.

$\square$

Therefore, in order to find the $A_{\cdot j}$'s that have the highest inner products with $A\mathbf{x}^{(t-1)} - \mathbf{y}$ in absolute value, we simply combine the set of $A_{\cdot j}$'s whose inner products with $A\mathbf{x}^{(t-1)} - \mathbf{y}$ are the highest and the set of $A_{\cdot j}$'s whose inner products with $\mathbf{y} - A\mathbf{x}^{(t-1)}$ are the highest and take the top half.

We now focus on how we can find the $A_{\cdot j}$'s whose inner products with $A\mathbf{x}^{(t-1)} - \mathbf{y}$ are the highest, since the procedure for finding those with $\mathbf{y} - A\mathbf{x}^{(t-1)}$ is the same.

Consider the Euclidean distance between $A_{\cdot j}$ and $A\mathbf{x}^{(t-1)} - \mathbf{y}$, which can be rewritten as:

$$\left\| A_{\cdot j} - \left( A\mathbf{x}^{(t-1)} - \mathbf{y} \right) \right\|_2 = \sqrt{\left( A_{\cdot j} - \left( A\mathbf{x}^{(t-1)} - \mathbf{y} \right) \right)^T \left( A_{\cdot j} - \left( A\mathbf{x}^{(t-1)} - \mathbf{y} \right) \right)}$$

$$= \sqrt{A_{\cdot j}^T A_{\cdot j} - 2 A_{\cdot j}^T \left( A\mathbf{x}^{(t-1)} - \mathbf{y} \right) + \left( A\mathbf{x}^{(t-1)} - \mathbf{y} \right)^T \left( A\mathbf{x}^{(t-1)} - \mathbf{y} \right)}$$

$$= \sqrt{\left\| A_{\cdot j} \right\|_2^2 - 2 \langle A_{\cdot j}, A\mathbf{x}^{(t-1)} - \mathbf{y} \rangle + \left\| A\mathbf{x}^{(t-1)} - \mathbf{y} \right\|_2^2}$$

Because $A$ is column-normalized, $\left\| A_{\cdot j} \right\|_2 = 1$ for all $j$. So,

$$\left\| A_{\cdot j} - \left( A\mathbf{x}^{(t-1)} - \mathbf{y} \right) \right\|_2 = \sqrt{1 - 2 \langle A_{\cdot j}, A\mathbf{x}^{(t-1)} - \mathbf{y} \rangle + \left\| A\mathbf{x}^{(t-1)} - \mathbf{y} \right\|_2^2}$$

Define $\phi(\mathbf{v}) = \langle \mathbf{v}, A\mathbf{x}^{(t-1)} - \mathbf{y} \rangle$ and $\psi(u) = \sqrt{1 - 2u + \left\| A\mathbf{x}^{(t-1)} - \mathbf{y} \right\|_2^2}$. Since $\psi$ is strictly decreasing in $u$, if $\phi(\mathbf{v}_1) \, \text{¿} \, \phi(\mathbf{v}_2)$, $\psi(\phi(\mathbf{v}_1)) < \psi(\phi(\mathbf{v}_2))$. In other words, a vector that achieves a higher inner product with $A\mathbf{x}^{(t-1)} - \mathbf{y}$ must be closer to $A\mathbf{x}^{(t-1)} - \mathbf{y}$ in Euclidean distance. Therefore, finding the $A_{\cdot j}$'s that attain the highest inner products with $A\mathbf{x}^{(t-1)} - \mathbf{y}$ is equivalent to finding the $A_{\cdot j}$'s that are closest to $A\mathbf{x}^{(t-1)} - \mathbf{y}$ in Euclidean distance.

The resulting algorithm is concretely stated in Algorithm 2.

---

**Algorithm 2** Accelerated Compressive Sensing Matching Pursuit (Accelerated CoSaMP)

---

**Require:** Column-normalized measurement matrix $A$, observed measurements vector $\mathbf{y}$ and sparsity level $k$

   $\mathbf{x}^{(0)} \leftarrow \mathbf{0}$
   $S \leftarrow \emptyset$
   Construct nearest neighbour search database $\mathcal{D}$ consisting of the vectors $\{A_{\cdot j}\}_{j=1}^n$
   **for** $t = 1$ **to** $T$ **do**
      $U_+ \leftarrow$ indices of $2k$ closest vectors in $\mathcal{D}$ to $A\mathbf{x}^{(t-1)} - \mathbf{y}$
      $V_+ \leftarrow$ absolute values of inner products between vectors indexed by $U_+$ and $A\mathbf{x}^{(t-1)} - \mathbf{y}$
      $U_- \leftarrow$ indices of $2k$ closest vectors in $\mathcal{D}$ to $\mathbf{y} - A\mathbf{x}^{(t-1)}$
      $V_- \leftarrow$ absolute values of inner products between vectors indexed by $U_-$ and $\mathbf{y} - A\mathbf{x}^{(t-1)}$
      $U \leftarrow$ indices of the vectors that the $2k$ largest elements in $V_+ \cup V_-$ correspond to
      $\tilde{S} \leftarrow U \cup S$
      $\tilde{\mathbf{x}}^{(t)} \Big|_{\tilde{S}} \leftarrow \left( \left( A|_{\cdot \tilde{S}} \right)^T A|_{\cdot \tilde{S}} \right)^{-1} A|_{\cdot \tilde{S}}^T \mathbf{y}$
      $\tilde{\mathbf{x}}^{(t)} \Big|_{\tilde{S}^c} \leftarrow \mathbf{0}$
      $S \leftarrow$ indices of the $k$ largest elements of $\tilde{\mathbf{x}}^{(t)}$ in magnitude
      $\mathbf{x}^{(t)} \Big|_S \leftarrow \tilde{\mathbf{x}}^{(t)} \Big|_S$
      $\mathbf{x}^{(t)} \Big|_{S^c} \leftarrow \mathbf{0}$
   **end for**
   **return** $\mathbf{x}^{(T)}$

---

Similar techniques can be applied to AIHT. The most significant difference is that the shrinkage operator is applied after a gradient descent step. Hence, *both* $\mathbf{x}^{(t-1)}$ and $A^T \left( A\mathbf{x}^{(t-1)} - \mathbf{y} \right)$ determine which elements will be zeroed out by the thresholding. Therefore, the coordinates corresponding to the columns of $A$ that achieve high inner products will not necessarily survive the thresholding, and the coordinates associated with low inner products may not be non-zero after thresholding. To apply the proposed technique in this case, we observe that $\mathbf{x}^{(t-1)}$ is sparse and simply evaluate the gradient descent step on the support of $\mathbf{x}^{(t-1)}$ and the nearest neighbours of $\pm \left( A\mathbf{x}^{(t-1)} - \mathbf{y} \right)$. The precise algorithm is stated in Algorithm 4 in the supplementary material.

## 4 SUFFICIENT CONDITION FOR FAST RECOVERY

Since our approach uses nearest neighbour search to accelerate sparse recovery, the amount of speedup we can obtain depends on the fundamental difficulty of the underlying nearest neighbour search problem. For the problem of exact nearest neighbour search, one standard way of characterizing of the difficulty is the *intrinsic dimensionality* (Karger & Ruhl, 2002; Dasgupta & Sinha, 2015), which is defined as follows:

**Definition 1.** *Given a dataset $D \subseteq \mathbb{R}^d$, let $B_p(r)$ be the set of points in $D$ that are within a closed Euclidean ball of radius $r$ around a point $p \in \mathbb{R}^d$. A dataset $D$ has expansion dimension $(\tau, d')$ if for all $r > 0, \alpha > 1$ and $p$ such that $|B_p(r)| \geq \tau, |B_p(\alpha r)| \leq \alpha^{d'} |B_p(r)|$. The quantity $d'$ is known as the intrinsic dimensionality.*

Intuitively, the intrinsic dimensionality characterizes how many close calls there could be when one tries to find the nearest neighbours. If we consider a ball around a query that contains a single data point (which is its nearest neighbour) and double the radius of the ball, then there could be as many as $2^{d'}$ data points inside the ball that are all at most a factor of 2 more distant from the query than the true nearest neighbour. Surprisingly, even though the number of points inside a ball grows exponentially in $d'$, there is a randomized algorithm that can find the nearest neighbours in time sublinear in $d'$ (Li & Malik, 2017).

We now derive a sufficient condition we would need to perform fast recovery. In our setting, nearest neighbour search is used to quickly the find the columns of $A$ that have the highest absolute inner products with $\left(A\mathbf{x}^{(t-1)} - \mathbf{y}\right)$. So, it would be ideal if we can find a sufficient condition in terms of inner products. As mentioned above, we will assume the columns of $A$ are normalized, which means all data points are unit vectors. We can also assume without loss of generality that query is normalized, since we can divide it by any constant without changing the rankings of the data points in terms of their inner products. For convenience, we will use $\mathbf{q}$ to denote the query and $\mathbf{p}^{(i)}$ to denote the data point with the $i$th highest inner product with $\mathbf{q}$, or equivalently, the $i$th shortest Euclidean distance to $\mathbf{q}$, since $\left\|\mathbf{p}^{(i)} - \mathbf{q}\right\|_2 = \sqrt{2 - 2\langle\mathbf{p}^{(i)}, \mathbf{q}\rangle}$.

First, we write down an equivalent statement to the definition in terms of $\mathbf{p}^{(i)}$'s and $\mathbf{q}$:

$$\forall j \geq \tau \; \forall i \geq \lfloor\alpha^{d'} j\rfloor + 1 \; \left\|\mathbf{p}^{(i)} - \mathbf{q}\right\|_2 \geq \alpha \left\|\mathbf{p}^{(j+1)} - \mathbf{q}\right\|_2$$

Then if we substitute $\sqrt{2 - 2\langle\mathbf{p}^{(i)}, \mathbf{q}\rangle}$ for $\left\|\mathbf{p}^{(i)} - \mathbf{q}\right\|_2$ and simplify, we get:

$$\forall j \geq \tau \; \forall i \geq \lfloor\alpha^{d'} j\rfloor + 1 \; \langle\mathbf{p}^{(i)}, \mathbf{q}\rangle \leq (1 - \alpha) + \alpha\langle\mathbf{p}^{(j+1)}, \mathbf{q}\rangle$$

We now simplify this expression condition by eliminating the variable $j$; we do so by finding the value of $j$ the results in the tightest inequality. Since $j$ appears on the right-hand side, we'd like to find the value of $j$ that results in the smallest $\langle\mathbf{p}^{(j+1)}, \mathbf{q}\rangle$, which happens when $j$ is large by definition of $\mathbf{p}^{(j+1)}$. At the same time, we need to make sure that $j$ is small enough so that $i \geq \lfloor\alpha^{d'} j\rfloor + 1$. This implies that $j < \alpha^{-d'} i$; because $j$ must be an integer, the inequality is the tightest when $j = \lfloor\alpha^{-d'} i\rfloor$. Hence, the condition simplifies to:

$$\forall i \geq \lfloor\alpha^{d'} \tau\rfloor + 1 \; \langle\mathbf{p}^{(i)}, \mathbf{q}\rangle \leq (1 - \alpha) + \alpha\langle\mathbf{p}^{(\lfloor(i/\alpha^{d'})+1\rfloor)}, \mathbf{q}\rangle$$

We now use this condition to derive the running time of Accelerated AIHT. For any $\mathbf{v}$, if we now let $A_{\cdot(i)}$ denote the $i$th column of $A$ with the highest inner product with $\mathbf{v}$, then define $d_0$ to be the smallest number such that $\forall\mathbf{v} \; \forall\alpha > 1 \; \forall i \geq \lfloor 2\alpha^{d_0} k + 1\rfloor \; \langle A_{\cdot(i)}, \mathbf{v}\rangle \leq (1 - \alpha) + \alpha\langle A_{\cdot(\lfloor(i/\alpha^{d_0})+1\rfloor)}, \mathbf{v}\rangle$, where $k$ is the target sparsity level. Then the running time of each iteration of Accelerated AIHT is:

- Finding $2k$-nearest neighbours using Prioritized DCI (Li & Malik, 2017): $O(mk \max(\log(n/2k), (n/2k)^{1-H/d_0}) + Hk \log H \left(\max(\log(n/2k), (n/2k)^{1-1/d_0})\right))$, where $H \geq 1 \in \mathbb{Z}$ is a free parameter chosen by the user
- Computing union of support and nearest neighbours: $O(k)$
- Taking gradient descent step: $O(mk)$

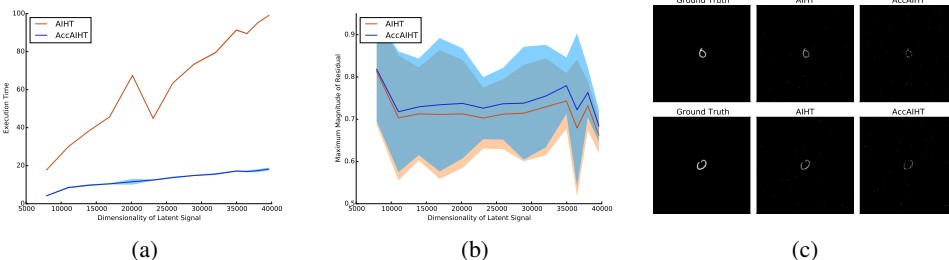

Figure 1: Performance of vanilla AIHT and accelerated AIHT on the image recovery task; (a) shows a comparison of the execution time at various sizes of images (lower is better), (b) shows a comparison of the magnitude of residuals (lower is better), and (c) visualizes the recovered images and the ground truth images.

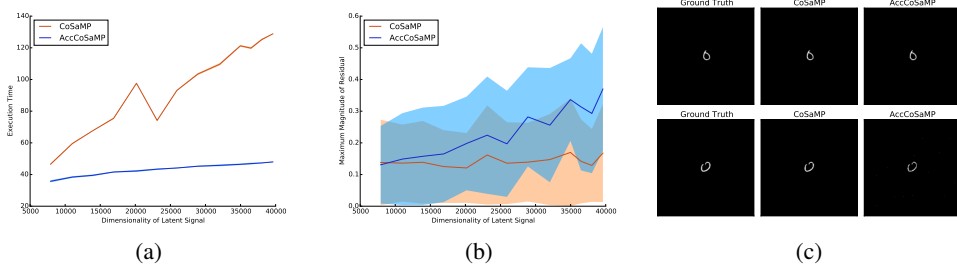

Figure 2: Performance of vanilla CoSaMP and accelerated CoSaMP on the image recovery task; (a) shows a comparison of the execution time at various sizes of images (lower is better), (b) shows a comparison of the magnitude of residuals (lower is better), and (c) visualizes the recovered images and the ground truth images.

- Taking the $k$ largest elements of $\mathbf{z}^{(t)}$: $O(k)$

So, the overall running time per iteration is $O(mk \max(\log(n/2k), (n/2k)^{1-H/d_0})$ $+$ $Hk \log H \left( \max(\log(n/2k), (n/2k)^{1-1/d_0})\right))$. Notice that it is sublinear in the signal length $n$. Note that as long as $d_0$ is relatively small, then this acceleration scheme could provide a significant reduction in the dependence of time complexity on $n$ relative to the vanilla version. The recovery guarantees is inherited from AIHT, since Prioritized DCI is guaranteed to return the correct set of nearest neighbours with high probability.

## 5 CONVERGENCE ANALYSIS

**Theorem 1.** *Denote $\tilde{\mathbf{x}}^{(t)}$ are the iterates generated from the original AIHT. Suppose $\tilde{\mathbf{x}}^{(t)}$ converges linearly to a $k$ sparse signal $\mathbf{x}^* \neq 0$ with rate $c$. And for any vector $\mathbf{v}$, the probability that the largest $k$ elements in $\mathbf{A}^T \mathbf{v}$ did not match the largest $k$ elements using DCI is at most $\epsilon$. Then with probability at least $1 - \epsilon^{\frac{\log |\mathbf{x}^*_{[k]}| - \log \|\mathbf{x}^{(0)} - \mathbf{x}^*\|}{\log c}}$, the iterates $\mathbf{x}^{(t)}$ generated from accelerated AIHT will be the same as $\tilde{\mathbf{x}}^{(t)}$, i.e., $\mathbf{x}^{(t)} = \tilde{\mathbf{x}}^{(t)}$ for any $i \in \mathbb{N}$.*

*Proof.* Because $\tilde{\mathbf{x}}^{(t)}$ converges linearly to $\mathbf{x}^*$, we have

$$\|\tilde{\mathbf{x}}^{(t)} - \mathbf{x}^*\|_2 \leq c^t \|\tilde{\mathbf{x}}^{(t)} - \mathbf{x}^*\|_2.$$

Because $\mathbf{x}^*$ is $k$ sparse, the $k$-th largest element in $\mathbf{x}^*$, denoted as $\mathbf{x}^*_{[k]}$, must be nonzero, i.e. $|\mathbf{x}^*_{[k]}| > 0$. Combining these two facts, we know that after finite steps, the support of $\tilde{\mathbf{x}}^{(t)}$ will not change. To be more specific, if $t > \frac{\log |\mathbf{x}^*_{[k]}| - \log \|\mathbf{x}^{(0)} - \mathbf{x}^*\|}{\log c}$, then $\|\tilde{\mathbf{x}}^{(t)} - \mathbf{x}^*\|_2 \leq c^t \|\mathbf{x}^{(0)} - \mathbf{x}^*\|_2 \leq |\mathbf{x}^*_{[k]}|$. If the

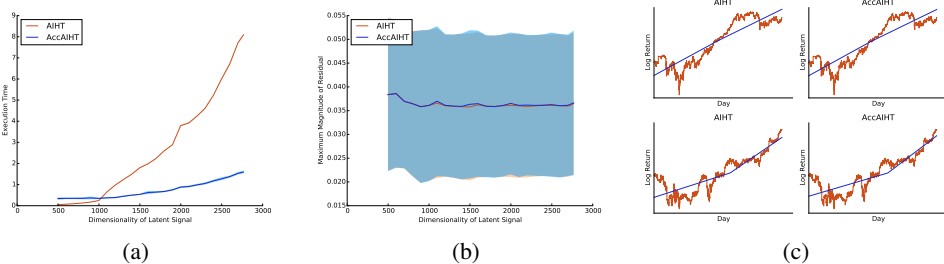

Figure 3: Performance of vanilla AIHT and accelerated AIHT on the trend detection task; (a) shows a comparison of the execution time at various resolutions of the time series (lower is better), (b) shows a comparison of the magnitude of residuals (lower is better), and (c) visualizes the extracted trends.

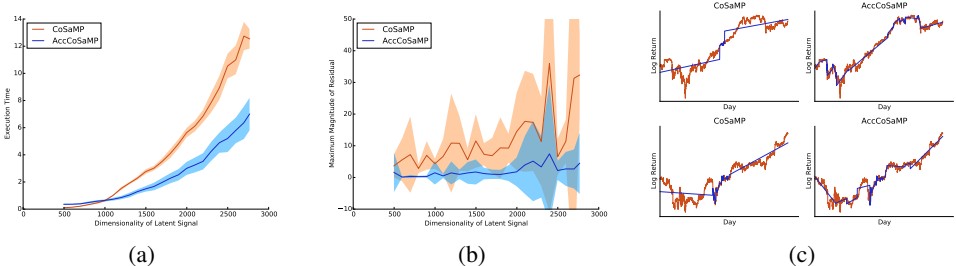

Figure 4: Performance of vanilla CoSaMP and accelerated CoSaMP on the trend detection task; (a) shows a comparison of the execution time at various resolutions of the time series (lower is better), (b) shows a comparison of the magnitude of residuals (lower is better), and (c) visualizes the extracted trends.

support of $\tilde{\mathbf{x}}^{(t)}$ does not contain the support of $\mathbf{x}^*$, their distance will be at least $|\mathbf{x}^*_{[k]}|$, which would contradict the fact that their distance is smaller than $|\mathbf{x}^*_{[k]}|$. Thus, the support of $\tilde{\mathbf{x}}^{(t)}$ must contain the support of $\mathbf{x}^*$ for any $t > \frac{\log |\mathbf{x}^*_{[k]}| - \log \|\mathbf{x}^{(0)} - \mathbf{x}^*\|}{\log c}$.

As for accelerated AIHT, the probability that $\mathbf{x}^{(t)} = \tilde{\mathbf{x}}^{(t)}$ for all $t \leq \frac{\log |\mathbf{x}^*_{[k]}| - \log \|\mathbf{x}^{(0)} - \mathbf{x}^*\|}{\log c}$ is at least $1 - \epsilon \frac{\log |\mathbf{x}^*_{[k]}| - \log \|\mathbf{x}^{(0)} - \mathbf{x}^*\|}{\log c}$. Now we will prove if that happens, iterates in accelerated AIHT will be the same as the iterates in AIHT for all $t$. We will prove by induction: if $\mathbf{x}^{(t)} = \tilde{\mathbf{x}}^{(t)}$ for all $t \leq t_0$ where $t_0 > \frac{\log |\mathbf{x}^*_{[k]}| - \log \|\mathbf{x}^{(0)} - \mathbf{x}^*\|}{\log c}$, then for $t = t_0 + 1$, as we discussed before, the support of $\tilde{\mathbf{x}}^{(t_0+1)}$ is the same as the support of $\tilde{\mathbf{x}}^{(t_0)}$. On the other hand, the support of $\tilde{\mathbf{x}}^{(t_0+1)}$ is the set of $k$ largest elements in $A^T(\mathbf{y} - A\mathbf{x}^{(t_0)})$. In the accelerated AIHT, the support of $\mathbf{x}^{(t_0+1)}$ is the same as the set of $k$ largest elements in $A^T_{\cdot S}(\mathbf{y} - A\mathbf{x}^{(t_0)})$ where $S$ is the union of the support of $\mathbf{x}^{(t_0)}$ and the set returned by DCI. Because that set always includes the support of $\mathbf{x}^{(t_0)}$, we know that the support of $\mathbf{x}^{(t_0+1)}$ will be the same as the support of $\mathbf{x}^{(t_0)}$. Therefore, we know that $\mathbf{x}^{(t_0+1)} = \tilde{\mathbf{x}}^{(t_0+1)}$. Thus, by induction, we know the conclusion holds. $\square$

## 6 EXPERIMENTS

We conduct experiments on real data using CoSaMP and AIHT and compare their performance to the accelerated versions developed in this paper. We consider two tasks, image recovery and trend detection in time series. In image recovery, we represent a sparse image using a randomly generated coding scheme, which each element of the code is a linear combination of all pixel values, where the

coefficients are randomly drawn i.i.d. from a standard Gaussian. The length of the code is much less than the number of pixels in the image. The goal is to reconstruct the original sparse image from the code. In trend detection, we take naturally occurring time series data and try to represent them parsimoniously as piecewise linear functions. The goal is to use as few linear pieces as possible, while approximating the original time series well.

We now present the concrete formulations of these tasks as sparse recovery problems. In image recovery, $A$ is an $m \times n$ matrix, where each entry is drawn randomly from a Gaussian. (Once generated, this matrix is fixed for all images of the same size.) Each row of $A$ represents a particular way of computing an element in the code, and each column corresponds to a pixel in the image. $\mathbf{x}$ is the image and $\mathbf{y}$ is the code. The image $\mathbf{x}$ is assumed to be sparse; in our experiments, we took MNIST digits (which generally have few white pixels) and padded them along the sides to obtain high-resolution images. The value of $m$ we used was 1800 and the value of $n$ ranged from 7000 to 40000.

In trend detection, $A$ is an $m \times n$ matrix, where each row is a discretization of the function $f(x) = \max(x - a, 0)$ at uniformly spaced values of $x$. Different rows have different horizontal shifts $a$. $m$ is always $n - 2$ in our case, since each subsequent row shifts the preceding row by one element. $\mathbf{x}$ is the coefficients on each of the linear pieces and $\mathbf{y}$ is the time series we would like to explain. We used the daily log closing prices of stocks traded on U.S. stock exchanges from 2007 to 2017 as our source of time series data. To get different resolutions of the data, we performed bilinear downsampling.

As shown in Figures 1 and 2, the accelerated versions of both AIHT and CoSaMP are much faster than the vanilla versions, while achieving comparable levels of accuracy. The speedup is more significant for AIHT because it only needs to perform a gradient step after thresholding, which is computationally inexpensive, whereas CoSaMP needs to perform least squares on the new support, which requires a comparable computational cost as finding the support.

As shown in Figures 3 and 4, the accelerated versions of both AIHT and CoSaMP are faster than the vanilla versions except in the very low-dimensional regime. Surprisingly, the accelerated version of CoSaMP actually achieves better accuracy than the vanilla version. We conjecture this is because the measurement matrix $A$ in the case of trend detection is much more structured and so the problem is more ill-conditioned, thereby making the randomness in the nearest neighbours search beneficial.

## 7 CONCLUSION

We presented a generic way of accelerating various sparse recovery algorithms, including CoSaMP and AIHT, and showed a sufficient condition under which acceleration is possible practically for free without sacrificing recovery guarantees. We also presented experiments on real world data, which shows that our algorithms achieve significant speedups over the vanilla versions.

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

# 8 SUPPLEMENTARY MATERIAL

---

**Algorithm 3** Adaptive Iterative Hard Thresholding (AIHT)

---

**Require:** Measurement matrix $A$, observed measurements vector $\mathbf{y}$, sparsity level $k$ and step size $\eta$

  $\mathbf{x}^{(0)} \leftarrow \mathbf{0}$

  **for** $t = 1$ **to** $T$ **do**

    $\mathbf{z}^{(t)} \leftarrow \mathbf{x}^{(t-1)} - \eta A^T \left( A\mathbf{x}^{(t-1)} - \mathbf{y} \right)$

    $S \leftarrow$ indices of the $k$ largest elements of $\mathbf{z}^{(t)}$ in magnitude

    $\mathbf{x}^{(t)}\big|_S \leftarrow \mathbf{z}^{(t)}\big|_S$

    $\mathbf{x}^{(t)}\big|_{S^c} \leftarrow \mathbf{0}$

  **end for**

  **return** $\mathbf{x}^{(T)}$

---

---

**Algorithm 4** Accelerated Adaptive Iterative Hard Thresholding (Accelerated AIHT)

---

**Require:** Column-normalized measurement matrix $A$, observed measurements vector $\mathbf{y}$, sparsity level $k$ and step size $\eta$

  $\mathbf{x}^{(0)} \leftarrow \mathbf{0}$

  Construct nearest neighbour search database $\mathcal{D}$ consisting of the vectors $\{A_{\cdot j}\}_{j=1}^n$

  **for** $t = 1$ **to** $T$ **do**

    $\tilde{S}_+ \leftarrow$ indices of $\left| \mathrm{supp}(\mathbf{x}^{(t-1)}) \right| + k$ closest vectors in $\mathcal{D}$ to $A\mathbf{x}^{(t-1)} - \mathbf{y}$

    $\tilde{S}_- \leftarrow$ indices of $\left| \mathrm{supp}(\mathbf{x}^{(t-1)}) \right| + k$ closest vectors in $\mathcal{D}$ to $\mathbf{y} - A\mathbf{x}^{(t-1)}$

    $\tilde{S} \leftarrow \tilde{S}_+ \cup \tilde{S}_- \cup \mathrm{supp}(\mathbf{x}^{(t-1)})$

    $\mathbf{z}^{(t)}\big|_{\tilde{S}} \leftarrow \mathbf{x}^{(t-1)}\big|_{\tilde{S}} - \eta \left( A|_{\cdot \tilde{S}} \right)^T \left( A\mathbf{x}^{(t-1)} - \mathbf{y} \right)$

    $\mathbf{z}^{(t)}\big|_{\tilde{S}^c} \leftarrow \mathbf{0}$

    $S \leftarrow$ indices of the $k$ largest elements of $\mathbf{z}^{(t)}$ in magnitude

    $\mathbf{x}^{(t)}\big|_S \leftarrow \mathbf{z}^{(t)}\big|_S$

    $\mathbf{x}^{(t)}\big|_{S^c} \leftarrow \mathbf{0}$

  **end for**

  **return** $\mathbf{x}^{(T)}$

---

