# OpenReview forum: "Accelerated Sparse Recovery Under Structured Measurements"
_ICLR.cc/2019/Conference_

### Official Review · AnonReviewer2 · 2018-10-29
**Accelerated Greedy Sparse Recovery Review**

**Rating:** 5
**Confidence:** 4

**Review:**

The paper proposes a greedy-like algorithm for sparse recovery that uses nearest neighbors algorithms to efficiently identify candidates for the support estimates obtained at each iteration of a greedy algorithm. It assumes that the norms of the columns of the matrix A are one to be able to change the project-and-sort step into a nearest neighbors search.

It is not clear what the value of Fact 1 is, given that none of the sparse recovery algorithms discussed here actually performs ell0 norm minimization. Additionally, it is common in theoretical analysis of sparse recovery to assume that the columns of the matrix A have unit norm. In fact, the RIP implies that the columns of the matrix must have norm within delta of 1. Nonetheless, it would be useful to have a discussion of the effect that having non-unit column norms would have on the proposed approach.

Similarly, Fact 2 is almost self-evident; I suggest to discard the proof.

The equivalence of Definition 1 and the statement involving ps and qs needs to be shown more clearly. The statement in Definition 1 is given in terms of distances (ball radiuses), not counts of neighbors.

I suggest swapping the use of CoSaMP and AIHT - the theoretical results of the paper refer to AIHT, so it is not clear why the algorithm itself is relegated to the supplementary material.

It is not clear how d0 is to be computed to implement Accelerated AIHT.

For Theorem 1, the authors should comment on when the assumption "xtilde(t) converges linearly to a k-sparse signal with rate c".

In Figures 1 and 2, does "residual" refer to the difference between x and xtilde, or b and Axtilde?

Minor comments:
Typo in page 5 "¿"
Grammar error in page 6 "characterizing of the difficulty".

---

### Official Review · AnonReviewer3 · 2018-11-02
**This paper proposes a unified framework for speeding up sparse regression algorithms by adapting fast nearest-neighbour search algorithms for updating the support.**

**Rating:** 5
**Confidence:** 3

**Review:**

The paper is very well-written, readable, with the ideas and derivations clearly explained.

The literature review is comprehensive and informative. I do feel however that the review could be improved, for example, by discussing the recent papers by Chinmay Hegde and Piotr Indyk on "head" and "tail" approximate projections to speed up recovery algorithms. The problem under study is indeed important and the contribution is interesting.

My biggest concern is that the technical contribution is too modest. Theorem 1 serves more as a decorative technical result (the assumption "And for any vector v..." seems out of the blue and too convenient) and the paper does not answer the many questions that come to mind here. For example, what is the intrinsic dimension of common random measurement matrices? Or how do any wrongly detected nearest neighbours propagate through the iterations of the algorithm? How does the measurement noise change the intrinsic dimension? We should intuitively lose stability in return for faster recovery. How would this be quantified in what you've proposed.

---

### Official Review · AnonReviewer1 · 2018-11-02
**This paper shows how to accelerate certain popular sparse recovery approaches under certain conditions. However, the contributions seem to be incremental and it is unclear how the technique significantly advance the state of the art.**

**Rating:** 4
**Confidence:** 5

**Review:**

Clarity: Paper is generally well written; however, certain theoretical statements (e.g. Theorem 1) are not very precise.

Originality: Contribution seems to be incremental; the proposed method seems to be a straightforward concatenation of well-known existing results in sparse recovery and nearest-neighbor search.

Significance: Unclear whether the techniques significantly advance the state of the art.

Quality: Overall, I think this is a promising direction but the idea might not have fully fleshed out.

----
Summary:
the paper proposes a scheme to accelerate popular sparse recovery methods that rely on hard thresholding (specifically, CoSaMP and IHT, but presumably other similar methods can also be used here). The key idea is that if the measurement matrix is normalized, then the k-sparse thresholding of the gradient update can be viewed as solving a k-nearest neighbor problem. Therefore, one can presumably use fast k-NN methods instead of exact NN methods. Specifically the authors propose to use the prioritized DCI method of Li and Malik.

Pros:
reasonable idea to use fast (sublinear) NN techniques in the k-sparse projection step.

Cons:
* It appears that the running time improvement over the baseline IHT (which has Otilde(mn) complexity) heavily depends on the intrinsic dimensionality of A. However, the authors do not characterize this.
* The authors neglect to mention in the paper that prioritized DCI has a pre-processing time of O(mn), so the final algorithm isn't really asymptotically faster.
* I cannot parse Theorem 1 (especially, the second sentence). Is epsilon the failure probability of DCI?
* Experimental results are far too synthetic. In real-life problems k itself is big, so there may be other bottlenecks (least squares, gradient updates, etc) and not necessarily the hard thresholding step.

---

### Meta-Review · Area_Chair1 · 2018-12-17
**Effective acceleration technique for sparsity regularized regression, but not complete enough**

**Confidence:** 5
**Recommendation:** Reject

**Metareview:**

The main idea of this paper is to use nearest neighbor search to to accelerate iterative thresholding based sparse recovery algorithms. All reviewers were  underwhelmed by somewhat straightforward combination of  existing results in sparse recovery and nearest-neighbor search.  While the proposed method seems effective in practice, the paper has the feel of not being a fully publishable unit yet. Several technical questions were asked but no author feedback was provided to potentially lift this paper up.